# Cardiopulmonary Resuscitation-Related Head Trauma: A Case Report

**DOI:** 10.3390/reports6040050

**Published:** 2023-10-19

**Authors:** Ilina Brainova, Pavel Timonov, Antoaneta Fasova, Alexandar Alexandrov

**Affiliations:** 1Department of Forensic Medicine and Deontology, Medical Faculty, Medical University of Sofia, Zdrave 2 Str., 1431 Sofia, Bulgaria; sashko_forensic@yahoo.com; 2Medical Faculty, Medical University of Plovdiv, 4000 Plovdiv, Bulgaria; pavelttimonov@yahoo.fr (P.T.); antoaneta_ioan@yahoo.com (A.F.)

**Keywords:** cardiopulmonary resuscitation, resuscitation-related trauma, forensic examination, postoperative complications, head trauma

## Abstract

Introduction: In all cases of cardiac arrest, adequate cardiopulmonary resuscitation (CPR) performance is crucial for survival. There are differences between the performances of CPR in pediatric cases compared to CPR in adults. In all cases in which CPR is needed, there is a possibility for the occurrence of CPR-related traumatic injuries. Aims and methods: We used all available forensic examination methods in order to provide objective forensic investigation conclusions and feedback to clinicians. Results: We present an untypical case of head trauma with intracranial bleeding caused via CPR-related traumatic injury. Although it is not connected with the mechanism and genesis of death, it should be noted as being practically casuistic. The child had a severe congenital heart malformation. The surgical team decided that surgery was absolutely necessary. Complications developed in the postoperative period. On the second postoperative day, cardiac arrest occurred, CPR was performed, and the girl survived for 15 minuntil there was a second cardiac arrest. A second CPR was performed, but she died. During the forensic autopsy, with the exception of expected findings, head bruising and intracranial bleeding were registered. Conclusion: The forensic conclusion was that the head trauma was caused accidentally during CPR due to the non-voluntary impact of the head with respect to the background of anticoagulant therapy, which was one possible factor for the massive bleeding.

## 1. Introduction

In all cases of cardiac arrest, a key determinant of patient survival is high-quality cardiopulmonary resuscitation (CPR) [1]. For out-of-hospital cardiac arrest victims, a bystander performing CPR remains the most important factor with respect to saving their lives [2]. When manually performed, the delivery of effective chest compressions might be inconsistent and subject to fatigue, and it is practically challenging. It might be said that mechanical CPR devices provide an automated way to deliver high-quality CPR [1]. However, there is no evidence of improved patient outcomes in patients treated with mechanical CPR compared to those treated with manual CPR [1,3].

CPR-related injuries might be observed either in cases of fatal outcomes or in cases of survivors. It is stated that trauma is independently associated with older age, the male gender, bystander CPR, cardiac etiology, the duration of CPR, and a lack of defibrillation. The analyses of registered postmortem traumatic injuries showed the prevalence of thoracic injuries—rib(s) and sternum fractures and fewer liver injuries [4]. The incidence of CPR-related injuries is higher in the non-survivors of out-of-hospital cardiac arrests compared to cases in an emergency medical service [5]. According to Setälä et al., 2018, the most common injuries are multiple rib fractures (43%), with 22% of patients having more than eight fractured ribs [5]. Injuries of the abdominal internal organs and injuries related to airway management are rare. Both bone fractures and internal organ injuries are registered after cardiopulmonary resuscitation. They are logically associated with the performance of mechanical chest compressions and active decompressions during CPR [6]. There are slight differences between the cases of manual compression compared to mechanical CPR, but none of the CPR-related traumatic injuries (sternum and rib fractures) were causes of fatal outcomes in actuality [7]. Penov et al., 2022, reported the incidence of aortic rupture due to prolonged mechanical CPR with concomitant Impella, which means that even some fatal complications of CPR might be possible [8]. Ho et al., 2022, presented a case of aortic dissection that was most likely an iatrogenic injury due to mechanical cardiopulmonary resuscitation carried out using the Lund University Cardiopulmonary Assist System (LUCAS) [9].

It is essential to know that appropriate pediatric CPR differs from CPR in adults. This fact is explained by the differences in the anatomical and physiological features in children compared to adults. Another important note is that the pathogenesis of cardiac arrests and the most common rhythm disturbances are different in children—they rarely suffer from sudden ventricular fibrillation cardiac arrest due to coronary artery disease. Amongst the most common causes of cardiac arrest in children are different conditions that causean acute hypoxic state. Optimal chest compression-to-ventilation ratios for children are not known and depend on a variety of factors, such as the compression rate and the tidal volume [10].

No cases of head trauma and intracranial bleeding might be described as typical or expected as traumatic injuries related to CPR performance.

The main aim of this case presentation is to provide feedback to clinicians with respect to possible CPR-related traumatic injuries. In relation to other studies and case presentations on the topic, we could not find any similar case reported. Our purpose is to draw attention to such possible complications because of their rarity.

The forensic experts were given the medical record of the patient, which concerns the surgery, postoperative complications, and fatal outcome. A full forensic autopsy was performed, including gross external and internal examination, and samples for additional histological (internal organs) and toxicological analyses (blood samples) were obtained if needed.

An autopsy in forensic cases starts when basic information about the circumstances of the fatal outcome is presented to the forensic examiner. Such data are generally reported to the forensic expert of the investigation. Additional information might be given by the relatives or other witnesses before or after the examination. In the presented case, it was crucial for the experts to be acquainted with the medical records of the deceased patient. It was necessary because, in such cases, suspicions referring to the correct and adequate actions of medical practitioners may arise. We were provided with medical documentation that was deemed important by the investigators for the forensic examination and investigation.

In the present case, the external examination was performed in the Forensic Department because the fatal outcome occurred in a hospital and was monitored clinically. In this case, the development of postmortem changes, which was noted via observation, palpation, and dynamic examination, fully corresponded to the information on the timing of the fatal outcome, which was noted in medical records. There were signs of the surgery and a soft tissue injury of the head described in the case’s presentation. In this case, there was no need to perform additional sections. The state of development and features of internal organs were noted.

The questions that had to be answered were as follows: What was the cause of death? What was the cause of the head trauma? How was it related to the CPR performance and the fatal outcome?

## 2. Detailed Case Description

### 2.1. Clinical Presentation

We present a case of a child with severe heart malformation that was surgically treated, but she passed due to complications after the surgery.

The child was female. The pregnancy was pathological, and the child was the mother’s second pregnancy. The pregnancy was complicated by the mother’s thrombophilia and COVID-19 infection during the 3rd month of pregnancy. The delivery was surgical; the weight of the newborn was 2500 g. The neonatal period was complicated. Due toa heart murmur, the baby required consultation with a cardiologist on the first day of her life. A severe heart malformation (tetralogy of Fallot) was established—the right ventricle had a double exit, D-malposition of the big arteries, subaortic interventricular defect, and pulmonary valve stenosis. The mother said that since the age of 3 months, the baby became cyanotic more often. The child was placed under continuous cardiological clinical supervision since birth. There were data for one hypoxemic crisis provoked by physical effort, with 70% saturation up to the age of 8 months. During the last month of her life, she became more anxious, with several hypoxemic episodes occurring. This was the reason for her hospitalization. It was decided that a surgical correction was necessary. During the surgery, a double ventricle correction was attempted, but it appeared to be impossible;thus, a single ventricle correction was performed. The period after the surgery was complicated: 1. Glenn circulation was dysfunctional; 2. there were respiratory problems, and the patient was extubated and secondarily intubated again due to respiratory insufficiency. The patient was on anticoagulation medication (heparin, 10 units per kilogram), and the effect was satisfactory. The APTT ratio was used to monitor the effect of heparin use, and it was within the normal referent borders. On the second day after the surgery, at 12:20 p.m., she developed acute cardiac insufficiency with cardiac arrest. Cardiopulmonary resuscitation was successfully performed for 10 min, and the heart function was restored. At 12:45 p.m., another cardiac arrest occurred, and despite performing CPR, the patient passed.

### 2.2. Autopsy Findings

The autopsy was performed on the second day after the fatal outcome. The cadaver was admitted to the Department of Forensic Medicine and Deontology by the order of the investigation. The postmortem changes corresponded to the time interval after which the death was noted in the medical record. The baby was 9 months old. Externally, the child seemed normally developed. Her height was 70 cm, and her body weight was 8300 g. The fat tissue was normally developed and present in the normal predilection areas for the age of the baby. During the external examination, no gross outer malformations were noted.

Signs of the surgical intervention were registered—there was a surgical wound (cut) fixed with surgical stitches on the front side of the thoracic area and vertically positioned in the middle of the thorax; there were also three more surgical cuts fixed with stitches, which formed the drainages. Needle punctures were also present in the right subclavian area and in the area of the right wrist.

During the external examination, the forensic experts noted a purple-reddish bruise in the lower part of the left parietal area (Figure 1A). It was round to elliptical and 2/1.5 cm in size. No other external traumatic injuries were present.

The internal examination continued in the classical order—the cranial cavity was open first, followed by the abdominal and thoracic cavities. Over the inner surface of the skin and the subcutaneous soft tissues in the left parietal area, there was contusion. The bones of the skull were intact and 0.2–0.3 cm thick. Nevertheless, the opening of the cranial cavity revealed the presence of a massive subdural hematoma as thick as 0.5 cm over the left brain hemisphere (Figure 1B). The hemorrhage was present at the base of the hemisphere as well. Its greatest thickness corresponded to the localization of the bruise of the soft tissues in the parietal area. In the same anatomical location, subarachnoid hemorrhage also occurred and was observed to be up to 0.2 cm thick (Figure 1B). The weight of the brain was 990 g. There were no brain contusions—neither coup nor contrecoup. The gross examination of the brain showed signs of edema—the brain’s surface was flattened. There were no signs of herniation.

The examination of the thoracic and abdominal cavities corresponded to clinically registered severe cardiac malformations, which were surgically treated, and the structure of the other organ systems (Figure 2A,B). The weight of internal organs was measured as follows: heart—45 g; right lung—90 g; left lung—75 g; kidneys—30 g each; liver—320 g; spleen—30 g. The internal organs exhibited heavy and congested gross appearances. Ascites, at approximately 100 mLof fluid, was present in the abdominal cavity.

The blood samples were not examined. The decision for their examination depends on the needs and questions of the investigation. In this specific case, the investigators decided that they would not need toxicological testing, and the samples were preserved in the department for 6 months.

The histological examination showed normal histological structure, acute venous congestion in the internal organs, and lung and brain edema. The examination of the samples from the intracranial bleeding showed no signs of hematoma organization, which corresponded with fresh bleeding that occurred right before the fatal outcome.

## 3. Discussion

CPR is related to the incidence of injuries, some of which might even be referred to as typical. The most common CPR-related traumatic injuries are those of the chest wall. Other injuries, including upper airways and pulmonary and intra-abdominal injuries, are rarely observed [11]. Factors such as older age, male gender, and public location might be independently associated with CPR-related injuries [5]. According to Hellevuo et al., 2013, in adult patients, there is an association between rib(s) and sternum fractures and greater mean and peak compression depths [12]. It is established that there are no differences in the incidence of PCR-related injuries between the cases in which mechanical chest compressions with the LUCAS device are performed and manual chest compressions [13].

The autopsy finding of the head trauma was pretty disturbing and caused expert difficulties, especially because it was most probably caused during CPR. There were no data available for any impact on the head; no such accident was noted in the medical documents. During the performance of CPR, there are possible traumatic injuries that are accepted as “normal” or at least expected and possible, even when all the guidelines are complied with. Such traumatic injuries that are commonly and clinically observed in forensic practice are rib(s) and sternum fractures and contusions of the surrounding soft tissues. A head trauma does not fall into this characteristically observed complex. In the presented case, based on the morphology of the skin and soft tissue contusion and subdural and subarachnoid hemorrhages, it might be supposed that they all occurred as a result of a single impact in the lower part of the left parietal area. In clinical conditions, without any present data of violent actions towards the child, it might be assumed that there was an involuntary, accidental single impact, which most probably occurred during the moving of the child for the correct performance of CPR. If we accept this possibility, we should state that such an accidental impact should have been noted in the medical record of the patient. The severity of hemorrhages in the cranial cavity might be explained by the medication given to the patient—anticoagulants—and by the mechanical increase in pressure with respect to the head due to the mechanical cardiac massage. The cause of death written on the death certificate was “postoperative complication of the main clinical condition–congenital cardiac malformation”. Most probably, the cardiac arrest was due to Glenn circulation dysfunction and developed respiratory problems, which are registered in the medical record. The head trauma overlapped but did not result in the fatal outcome. Nevertheless, we cannot exactly state its role in the development of the pathogenesis of the fatal outcome. We suggest that if there is causation between the trauma and the fatal outcome, it is partial causation. The authors could not find any other similar cases presented in the reviewed literature. This case is more of an exception than a commonly observed complication or CPR-related trauma, which explains why there are no similar cases described in the literature. The expert point of view and conclusion is that the leading and primary cause of death is severe heart malformation. We state and recommend that clinical doctors and nurses should become acquainted with the autopsy report (when appropriate, according to the investigation); in this manner, they can receive feedback, and the forensic expert of the investigation might be provided with some explanation of the dynamics of the deceased’s last moments. A recommendation could be given for the careful and alert manipulation of patients, especially at such a young age, when CPR is performed. The main limitation is that the mechanism and genesis of the trauma cannot be fully understood and explained with certainty. This is due to the lack of information and the absence of any noted trauma prior to the death in the medical records.

## 4. Conclusions

Based on the full analysis of the case, the cause of death is primarily the congenital severe cardiac malformation with secondarily developed complications that included the development of postoperative complications as well. The decision for surgical treatment and the approach were correct. The head trauma registered during the forensic autopsy was a result of an impact that was most probably accidental during the performance of CPR. The trauma itself cannot be accepted as a cause of death, but it should have been at least noted as an impact on the medical record of the patient. Even though accidents are sometimes possible in clinical conditions, they should be prevented where possible because they might lead to serious complications or, in this case, result in expert conclusion difficulties. The head trauma overlapped with the mechanism and genesis of the fatal outcome, and even if it was not fatal or the leading cause of death, we, as experts, were not able to exclude its presence or its possible role.

## Figures and Tables

**Figure 1 reports-06-00050-f001:**
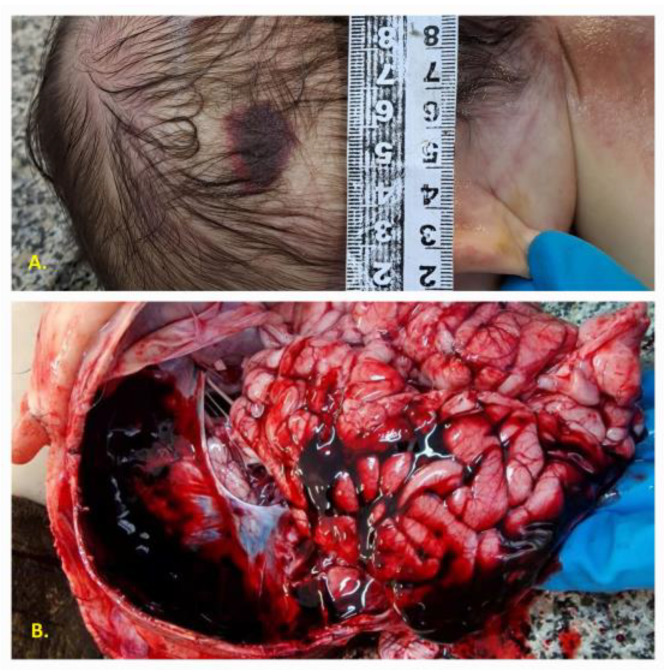
(**A**). Bruise in the left parietal area. (**B**). Intracranial bleeding—subdural and subarachnoid hemorrhages.

**Figure 2 reports-06-00050-f002:**
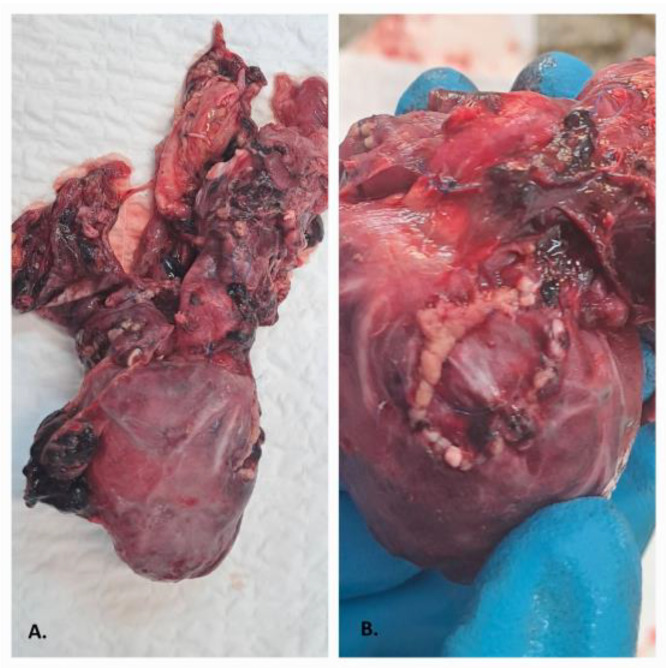
(**A**,**B**): The heart after the surgery (stitches are visible).

## Data Availability

The data presented in this study are available on request from the corresponding author. The data are not publicly available due to privacy.

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
