# Peer review of "Cardiopulmonary Resuscitation-Related Head Trauma: A Case Report"

_reports, 2023, doi:10.3390/reports6040050_

Round 1

Reviewer 1 Report

The manuscript

    The data you collected in this manuscript is good. Based on these data, you might have a good study result. Unfortunately, this manuscript does not reach the criteria for publication.

Overall:
1. There are many spelling errors in this manuscript. See guide for authors for a free grammar checker. E.g.……..etc. 

Title: - "Head Trauma Accidentally Caused during CPR: A Case Report." is a brief phrase describing the contents of the paper.

Abstract is often composed of five parts including Introduction, aim, methods, results, and conclusion. The background should be integrated into aim.: -
1. Introduction: relevant.

2. Aim of the study: please add the aims or purposes of the research and its relationship with other studies in the field.
3. Methods: Please add the study methodology that is used in this study in more detail.

4. The result: - please added results and should be presented with clarity and precision,
5. The conclusion section should answer the question you proposed in aim section.
6. The keywords are following the abstract, use about five to 10 key words do not mention in the title so please changes these words without any abbreviations.

Introduction
1. The introduction is ideally in structures.

Methods

1.The methods section should write in a way that everyone could repeat this study in the same manner.
2. You should add study questions, primary objectives, and secondary objectives if available.

Discussion

1.A discussion should offer a short overview of the results, and an in-depth discussion of the interpretation of them.
2. Do the authors have an explanation on why the results are different compared to other studies?

Limitations of the study: -

You should add.

Conclusion:
is logic in manner and relevant.

References:
1. Please use uniform references, when available with DOI.

2. Make sure update the old reference.

1. There are many spelling errors in this manuscript. See guide for authors for a free grammar checker. E.g.……..etc.  

Reviewer 2 Report

The manuscript describes a case of head trauma after performing CPR on a nine-month-old child. The case is interesting but raises important questions which had to be addressed.

My suggestions:
Lines 70-85 are irrelevant, therefore, should be deleted (if the authors feel necessary, reference to autopsy standards, like No99. European Council Recommendation should be included).
Line 69: the authors mention samples were obtained. Please indicate what type of samples were obtained.
There is no mentioning of histological and toxicological results. If toxicology was not made, then it should be indicated. Histology is, however essential. Please include the histological findings, with a particular focus on the injuries, especially the dating of these injuries; and also on the heart.
Line: please detail the anticoagulation medication, and also the coagulation parameters from the laboratory examination should be included (especially because these medications could play a role in the development of the subdural bleeding - as suggested in the discussion)
Line 108 and 109 describes the CPR. It is essential to describe it in detail with a timetable. To discuss the possible mechanism, it would be crucial to include information about where it was performed, what objects were present, and how the patient was moved before/during the CPR.
Line 115 is unnecessary.
Line 126-127: please include information wheter there were any other injuries present.
Line 131: remove "infiltration with blood". Contusion is enough. But the size and colour should be included.
Line 132: please include the thickness of the cranium.
Line 133: please include the size of the subdural hematoma (weight or volume)
Line 136: please include the extent and thickness of subarachnoidal haemorrhage
Figure 1: two pictures from the brain after the removal of the skull, before removal of arachnoid and after the removal of arachnoid should be included (one which shows the extent of subarachnoidal bleeding and another one which shows the brain surface)
There is no mention of any brain contusion found (coup-contracoup). What was the brain weight? Were there any signs of brain edema? Were there any signs of herniation?
Line 146: a little bit more detail of internal findings is necessary (organ weight, signs of heart failure...etc...).
What is missing from the discussion & conclusion:
Is there an explanation how and when the victim could suffer a head injury during CPR? Were there any legal consequences /medical malpractice/. If yes what, if not then why.
What was the cause of death (written on the death certificate)?
What was the expert's opinion about the possible role of the trauma in the outcome (direct or indirect causation, partial causation)
What was the possible cause of cardiac arrest?

It would be also interesting to see (like a review of literature) wheter there were any cases of severe head trauma suffered during CPR previously reported in the literature.

The manuscript is easy to read and understand, but there are some grammatical and stylistic issues. I recommend correcting it with a native english speaker.

Round 2

Reviewer 1 Report

the manuscript is certainly interesting, and the topic is very relevant. There are potentially very valuable lessons to learn from this manuscript. However, the revised manuscript is suitable for publication.

Title: -Case Report CPR-related head trauma: A Case Report

Reviewer 2 Report

Dear Author's!

Thank you for adding the missing information, i recommend the acceptance of the manuscript  in the present form.